# Agronomic and Metabolic Responses of *Citrus clementina* to Long-Term Irrigation with Saline Reclaimed Water as Abiotic Factor

**DOI:** 10.3390/ijms26073450

**Published:** 2025-04-07

**Authors:** David Auñón-Calles, María Pinciroli, Emilio Nicolás, Angel Gil-Izquierdo, José Antonio Gabaldón, María Puerto Sánchez-Iglesias, Angel Antonio Carbonell-Barrachina, Federico Ferreres, Carlos J. García, Cristina Romero-Trigueros

**Affiliations:** 1Molecular Recognition and Encapsulation Research Group (REM), Health Sciences Department, Universidad Católica de Murcia (UCAM), Campus de los Jerónimos 135, 30107 Guadalupe, Spain; daunon@ucam.edu (D.A.-C.); jagabaldon@ucam.edu (J.A.G.); fferreres@ucam.edu (F.F.); 2Research Group on Quality, Safety and Bioactivity of Plant Foods, Department of Food Science and Technology, CEBAS-CSIC, University Campus of Espinardo–Edif. 25, 30100 Murcia, Spain; pinciroli1@gmail.com (M.P.); mpsanchez@cebas.csic.es (M.P.S.-I.); cjgarcia@cebas.csic.es (C.J.G.); 3Cátedra de Bioquímica y Fitoquímica, Facultad de Ciencias Agrarias y Forestales, Universidad Nacional de la Plata, Calle 60 y 119, La Plata 1900, Buenos Aires, Argentina; 4Department of Irrigation, CEBAS-CSIC, University Campus of Espinardo–Edif. 25, 30100 Murcia, Spain; emilio@cebas.csic.es; 5Department of Agro-Food Technology, University Miguel Hernández-EPSO, Carretera de Beniel, km 3.2, 03312 Orihuela, Spain; angel.carbonell@umh.es

**Keywords:** salt stress, amino acids, phytohormones, reclaimed water, fruit quality, leaf nutrients, mandarin trees

## Abstract

The Panel on Climate Change has predicted an intensification of drought and heat waves. The aim of this study was to determine the physiological response of mandarin trees in a semi-arid area to the effects of a long period of irrigation with saline reclaimed water (RW) and freshwater (FW) in terms of leaf mineral constitution, free amino acids and phytohormone balance, and their influence on yield and fruit quality. Results showed that higher foliar levels of Cl^−^_,_ B, Li^+^, and Br^−^ were found in the RW treatment. In addition, fruit quality (juice content, soluble solid content, titratable acid, and maturity index) and yield (fruit weight and diameter) parameters and growth canopy were negatively affected by irrigation with RW. Regardless of the treatments, L-alanine (Ala) and proline were the most abundant amino acids, with Ala being described as a majority for the first time in the literature. Concretely, in FW, the total amino acid content was twice as high as the concentration in RW (51,359.46 and 23,833.31 ng g^−1^, respectively). The most abundant hormones were 1-Aminocyclopropane-1-carboxylic acid and *trans*-zeatin in both treatments. The saline stress response would be reflected in the higher concentration of salicylic and abscisic acids in the leaves of RW trees. In view of the high correlations found in a simplified correlation matrix of (i) Ala with the canopy growth and (ii) the salicylic acid (SA) with most of the evaluated agrometabolic parameters, it can be concluded that the exogenous application of the Ala and SA would increase tree size and could mitigate the effects of salt stress, respectively. However, these treatments could be completed with the external application of ACC since this phytohormone presents the lowest parameter during treatment with RW.

## 1. Introduction

The Intergovernmental Panel on Climate Change has predicted an intensification of drought and heat waves in the years to come that will cause important economic losses due to a reduction in the yield and quality of crops and will represent a threat to food safety. However, irrespective of what the climate does in the future, globalization and socio-economic changes are the major drivers for increases in water demand and threats to water safety, as exemplified by the burgeoning economies of the BRIC (Brazil, Russia, India, China, and South Africa) and MINT (Mexico, Indonesia, Nigeria, and Turkey) countries and the large population increases and economic growth seen in many developing countries [1]. Nowadays, there is an urgent need to improve water management by developing new water-saving strategies, especially in agriculture [2]. The worldwide use of reclaimed water (RW) is developing very rapidly, mainly in arid and semi-arid countries [3]. It is predicted that by 2030, 1.66% of the world’s total water will be reclaimed water [4], and agricultural irrigation is the main application for water reuse worldwide. In fact, the Water Reuse Regulation, which sets uniform minimum water quality requirements for the safe reuse of treated urban wastewater, has recently come into force in recognition of the growing importance of reused water in agriculture [5].

RW has excellent potential to become, with appropriate management, a valuable irrigation water source [6,7]. It influences plant nutrition positively by rendering the concentration of macronutrients, i.e., N, P, and K^+^, closer to their optimum levels for plant growth, which might also reduce fertilizer application rates [8]. RW irrigation may also promote the activity of soil microbial communities in citrus crops [9] or microbial biomass [10] in accordance with the Green Deal European strategy to promote soil health [11]. However, inappropriate management of irrigation with RW can exacerbate problems of nutrient leaching [12] (Romero-Trigueros et al., 2014) or secondary salinization and soil degradation in the medium–long term and finally result in negative impacts on crop physiology, growth, yield, and quality [13]. Consequently, damage caused by salinity has been primarily associated with ion toxicity. It has been reported that above an electrical conductivity (CE) threshold value of 1.4 dS m^−1^, every 1 dS m^−1^ increase results in an average of 13% decrease in citrus yield [14,15]. Salinity adversely influences several aspects of citrus plant vegetative and reproductive growth. Among the physiological processes leading to a general reduction in growth, photosynthetic activity, transpiration, stomatal conductance, and even root hydraulic conductivity have been shown to be decreased by salinity [13]. Moreover, under these conditions, a wide range of plant nutritional deficiencies is also apparent [16]. High salinity in irrigation water has been reported to reduce flowering intensity, fruit set, number of fruits, and fruit growth. The relationship between salinity and yield can normally be expressed as a negative linear response function [15]. However, several authors have reported that fruits produced under moderate salinity have higher sweetness and flavor and firmer fruits [15]. Particularly, citrus are generally classified as “salt-intolerant” crops since irrigation with salinized water immediately arrests tree growth and negatively affects fruit quality more than in many other crops. For example, in Cleopatra mandarin, the threshold for salinity stress syndromes is 2.08 dS m^−1^. However, among the citrus species, mandarin is the best plant to cope with the negative impact of water deprivation. Therefore, it is the best for drought resistance: mandarins (*Citrus reticulata* spp.) > rangpur lime > rough lemon > sour orange > *Citrus macrophylla*; medium resistance: lemon > trifoliate orange > citrange hybrid > *Citrus chuana*; poor tolerance: sweet orange > *Citrus verrucose* > grapefruit [17].

All tree-specific traits controlling and influencing flowering, fruit development, abscission, and ripening are obvious major areas of research. This control involves adaptive metabolic changes signaled by different intervening hormones at different times. The hormonal responses depend on the genotype and stage of development of the plant at the time of stress, the duration and severity of the stress, and the environmental factors causing the stress. Plant tolerance also depends on the performance of plant metabolism. Most responses and adaptations seem to be regulated by hormone levels and changes in sensitivity to them. Phytohormones are generally considered to be part of the signal transmission mechanism produced by environmental stimuli [18]. For example, abscisic acid (ABA) is considered the main regulator of plant responses to salinity because it promotes growth reduction, leaf senescence, stomatal closure, osmotic adjustment, increased root conductance, and gene induction [18]. In mature citrus fruits, exogenous ethylene accelerates color breakthroughs in both chlorophyll degradation and carotenoid deposition. Leaf-derived cytokinins (zeatin) could play a role in the stimulation of bud growth in evergreen trees [19]. Certain amino acids are involved in hormone synthesis and in abiotic stress tolerance mechanisms. Methionine is an essential precursor in the biosynthesis of 1-aminocyclopropane-1-carboxylic acid (ACC) [20], leucine, and proline in the synthesis of gibberellins [21]. Alanine, arginine, and glycine are involved in chlorophyll synthesis, while lysine, glutamic acid, and glycine play a prominent role in plant resistance systems [22]. To date, there are few previous studies describing the relationship of irrigation with saline water and their metabolic behavior in terms of phytohormones profile [23,24]. To the best of our knowledge, this is the first work that seeks to associate metabolic changes in terms of phytohormones and amino acids in mandarin trees irrigated with saline reclaimed water with respect to agronomical parameters.

We hypothesize that water quality may affect agronomic parameters in mandarin trees, and this may be justified through agrometabolic leaf composition.

## 2. Results and Discussion

In order to rule out any additional abiotic stresses that might occur during the experiment, the weather conditions were first analyzed. Then, the irrigation water, the nutritional status, the phytotoxic elements, the amino acids, and the hormones present in the leaves at the III stage of grain maturity (at the beginning of the third stage of rapid fruit growth) were analyzed. These parameters could influence the harvest at this phenological stage. Finally, the yield and the quality of the mandarin crop with and without salt stress (RW and FW, respectively) were evaluated.

### 2.1. Meteorological Conditions and Irrigation Water Analysis

Weather conditions were favorable for growing. Reference evapotranspiration (ET_0_) and rainfall were 1393.8 mm and 401.2 mm, respectively, during the year under review. This excluded other abiotic factors, which could cause stress during the study period. Figure 1 shows the meteorological records corresponding to the 15 days before and after the key dates: A (leaf sampling at the beginning of the third rapid fruit-growing phase) and B (harvest).

As can be seen in Table 1, FW had an electrical conductivity (EC_w_) of 1.38 dS m^−1^ and a sodium adsorption ratio (SAR_w_) around 3.35 meq L^−1^, while RW had higher values of both parameters (EC_w_ = 3.41 dS m^−1^ and SAR_w_ 6.09 meq L^−1^) than FW.

It has been reported that above a threshold EC_w_ value of 1.4 dS m^−1^, every 1 dS m^−1^ increase results in an average of 13% decrease in citrus yield [14,15]. The pH is slightly lower in RW than in FW. The concentrations of the tested ions Ca^2+^, Mg^+2^, K^+^, Na^+^, Cl^−^, NO_3_^−^, PO_4_^3−^, and SO_4_^2−^ are, on average, twice as high for RW as for FW. Even the concentration of B in RW was more than 3-fold higher than in FW (Table 1).

### 2.2. Response of Nutrient and Agrometabolites at Leaf Level to Saline-Reclaimed Water

#### 2.2.1. Nutritional Status

To find out the nutritional status of the mandarin tree at the beginning of stage III of fruit ripening, phytotoxic elements (Na^+^, Cl^−^, B, Li^+^, and Br^−^), macronutrients (N, P, K^+^, Ca^2+^, S, and Mg^2+^), and micronutrients (Cu, Zn^2+^, Mn, Fe, and Ni) concentrations at leaf level were measured for both irrigation treatments (FW and RW) (Table 1).

Despite the high salt concentration of the irrigation water used (RW), the Na^+^ concentration in mandarin leaves was 0.05% in both treatments. Boman [25] has found symptoms with concentrations between 0.10 and 0.25% dry wt. Romero-Trigueros et al. [26] identified Na concentrations above 0.1% as the phytotoxicity threshold for citrus yield reduction.

The Cl^−^ concentration of RW is more than twice that of FW. The mean values were 0.40 and 0.98% in the FW and RW, respectively. Ferguson and Grattan [27] suggested that leaf Cl^−^ becomes toxic in citrus trees when it reaches 0.7% dw. Boman [25] observed growth retardation at 1.5% leaf Cl^−^ and toxicity at 1% leaf concentration. In Romero-Trigueros et al. [26], 0.6% salt-induced phytotoxicity was identified as the threshold at which yield reduction occurs in citrus crops. The fact that the Carrizo citrus (*Citrus sinensis* [L.] Osb. × *Poncirus trifoliata* [L.]) rootstock does not exclude Cl^−^ ions may explain the high Cl^−^ content observed in mandarin leaves irrigated with RW [28]. In this sense, other authors [29] highlighted the importance of scions for salinity tolerance, specifically chlorides.

In our experiment, foliar concentrations of Na^+^ were similar in both treatments, contrary to what was observed by Pérez Pérez et al. [28]. This result was in agreement with that observed by Nicolás et al. [8] in mandarin trees during a series of six irrigation seasons with RW. According to the authors, this could be because good soil K^+^ supply may have reduced Na^+^ uptake by the tree, limiting transport of this phytotoxic ion from roots to leaves [8]. It can also be explained by the fact that citrange “Carrizo” (*Citrus sinensis* × *Poncirus trifoliata*) is a rootstock excluder of Na^+^. 

Higher levels of B, Li^+^, and Br^−^ were found in the leaves of trees irrigated with RW (Table 2). The B concentration in mandarin leaves irrigated with RW was twice as high as in the FW treatment. The values were 83.64 and 169.91 mg kg^−1^, respectively. The RW treatment exceeded the 100 mg kg^−1^ damage threshold established by Romero-Trigueros et al. [26]. B^3+^ uptake by plant roots was an active, energy-dependent transport through a high-affinity uptake system [30], implying additional energy consumption detrimental to plant growth potential. B^3+^ concentration affected the leaf mineral concentration of the trees, causing visible nutritional deficiencies. The Li^+^ content was 54% higher in the mandarin leaves irrigated with RW, with values of 6.81 and 10.48 mg kg^−1^ for the FW and the RW, respectively (Table 2). Li reduces plant growth by interfering with numerous physiological processes, altering plant metabolism, and reducing food quality [30]. However, only leaves containing more than 12 mg. kg^−1^ show symptoms of lithium toxicity [30]. Br^−^ values of 0.09 and 0.21 mg kg^−1^ were 133% higher for RW than for FW. Martin et al. [31] found that high concentrations of Br^−^ reduced growth.

Although RW had higher concentrations of the macroelements analyzed (total N, NO_3_^−^, PO_4_^3−^, and SO_4_^2−^), trees irrigated with RW did not show significantly higher levels of any of these ions at the leaf level, except S (Table 2). A possible competition between Cl^−^, N, and P uptake can be observed in citrus, according to Cámara et al. [32], due to trees irrigated with saline RW showing a tendency to reduce the leaf content of N and P. However, an increasing trend in N species has been observed (Table 2), which could be linked to a proper calculation of irrigation with RW since an excess of 2500 mm per year in such irrigation water could cause a decrease in nitrogen uptake efficiency in citrus [33] (Scholberg et al., 2002). The lower S^2−^ ion content in RW may explain the effects found on tree growth and crop yield. Recent studies have demonstrated that S is important for the proper growth, metabolic activities, and development of plants. S is one of the most essential macronutrients required by plants, as it is an important constituent of amino acids such as cysteine and methionine and in many metabolites like essential vitamins, sulfur esters, and sulfur derivatives [34].

No significant differences were observed among the treatments in the concentrations of the analyzed micronutrients (Table 2). The contents of Zn^+2^, Mn, and Fe in the FW and RW were lower than those observed by Kaur et al. [35] in the leaves of kinnow mandarin budded on the rootstock of Jatti Khatti (*C*. *jambhirilush*) (Table 2). The fact that they have different rootstocks may explain these differences in micronutrient concentrations.

The heavy metals analyzed in this experiment (Pb, Cd, Tl, Cr, Ti, As, and Cu) did not show any differences between the treatments, and there was no visible toxic damage.

#### 2.2.2. Amino Acid Metabolism

In biology, amino acids have vital roles in cell life. Eighteen free amino acids were identified in both irrigation treatments. In the analysis of each amino acid in particular (and as an average of the concentrations present in both treatments), the order of presence was as follows: Ala (21,899.15 ng g^−1^) ≥ Pro (8698.88 ng g^−1^) ≥ Tyr (2523.85 ng g^−1^) ≥ GABA (1114.31 ng g^−1^) ≥ Ser (863.31 ng g^−1^) ≥ Cys-Gly (785.72 ng g^−1^) ≥ Leu (374.96 ng g^−1^) ≥ Arg (372.11 ng g^−1^) ≥ Asn (255.18 ng g^−1^) ≥ Glu (193.94 ng g^−1^) ≥ Val (150.43 ng g^−1^) ≥ Met (103.88 ng g^−1^) ≥ Asp (81.22 ng g^−1^) ≥ Gly (64.53 ng g^−1^) ≥ Phe (38.49 ng g^−1^) ≥ Citrulline (36.87 ng g^−1^) ≥ Thr (23.75 ng g^−1^) ≥ His (15.81 ng g^−1^).

The first two (Ala and Pro) account for 85% and 73% of the total amino acid content in the leaf for the FW and RW treatments, respectively (Table 3). Ala is observed as the most abundant amino acid for the first time in the literature. According to Rabe and Lovatt [36], proline and serine were the most abundant, accounting for 58% of the total. Proline and serine are the most abundant free amino acids present in Satsuma leaves [37]. Special attention deserves this amino acid (Pro), as it presents a tendency to diminish after treatment with RW; this could be linked in Citrus to the exudation that has been previously described as a consequence of exposure to salt stress and as a form of rhizosphere communication [38] (Vives-Peris et al., 2017). Alanine ranked 5th in the amino acid content of young, fully expanded leaves of four citrus species (*Citrus limon* (L.) Burm. f. cv. rough lemon, *Poncirus trifoliata* (L.) Raf. cv. Australian trifoliate orange, and *P. trifoliata* x C. sinensis (L.) Osbeck cv. Carrizo and cv. Troyer citrange) irrigated with water of good agronomic quality.

RW had lower levels of all amino acids tested than FW, although, of the 18, only Ala, Ser, and Gly were significantly different, with 66.8%, 45.7%, and 55.8% reductions (Table 3). The higher levels of Ala, Ser, and Gly in the FW contributed to the maintenance of the better physiological condition of the plants in the absence of stress, as was observed in other works on this same experiment [26,40] due to these three amino acids being related to gas exchange parameters. Alanine promotes the photosynthetic process, generating more chlorophyll in leaves and consequently light use efficiency [22]; Ser plays an indispensable role in several cellular processes in one-carbon metabolism [41]; and Gly is responsible for growth stimulation [42].

The concentration of the remaining 15 amino acids did not change when the leaves were irrigated with RW.

#### 2.2.3. Individually Characterizing Phytohormones

Eleven phytohormones were quantified in average mature FW and RW mandarin leaves (Table 3). Analyzing each of the two treatment average concentrations, the following order of occurrence was observed: ACC (2114.41 ng g^−1^) ≥ tZ (930.04 ng g^−1^) ≥ SA (227.60 ng g^−1^) ≥ GA4 (58.11 ng g^−1^) ≥ JA (5.12 ng g^−1^) ≥ ABA (1.31 ng g^−1^) ≥ Zr (0.68 ng g^−1^) ≥ IPA (0.15 ng g^−1^) ≥ GA3 (0.07 ng g^−1^) and no presence of GA1 and AIA detected.

The most abundant hormones were ACC and tZ. They represented 93% and 89% of the total hormones in the FW and RW treatments, respectively.

ACC was the major phytohormone in the leaves of both treatments. RW had lower concentrations of ACC than FW. Methionine, an amino acid precursor of this hormone, is responsible for the low concentration of ACC observed in RW, which is in line with the reduction in this amino acid under such conditions (Table 3). Although this difference was not statistically significant, the concentration of ACC was 1.6 times lower in RW than in FW. S is also a precursor of methionine, so the low S content in RW (Table 2) may have had an influence on the methionine deficiency and, thus, the lower ACC content observed in RW.

The second most abundant hormone in both treatments was tZ; in this case, it was 62% higher in RW (Table 3) than in FW. The leaf-derived cytokinins (zeatin) could play a role in the stimulation of bud growth in evergreen trees; cytokinin active fractions corresponding to ribosyl zeatin, isopentenyl adenine, and cytokinin ribotides were present in all developmental stages [43].

The third hormone, ranked according to abundance, was salicylic acid (SA). SA alters key plant functions, including nutrient uptake, membrane functioning [44], water relations [45], stomatal functioning [46], inhibition of ethylene biosynthesis [47,48], and increased growth [23]. SA concentration in RW was 157% higher than the concentration in the leaves of control plants. In RW treatment, the high SA concentration could be related to the protective function of SA in membranes, which could increase plant tolerance to salt stress [49] by decreasing electrolyte loss and also with the exudate of this phytohormone in the root environment as a co-communication agent of the rhizosphere in citrus [38].

Similarly, SA levels in RW may have contributed to maintaining stable macro- and micronutrient concentrations in both treatments (Table 3), as it affects nutrient availability and mobility [50]. SA may also contribute to increasing levels of ABA (Table 3). ABA is a phytohormone that induces the synthesis of a wide range of anti-stress proteins and thus provides plant protection [51]. Exogenous foliar application of SA resulted in cell division in the root apical meristem, thereby increasing the growth and productivity of salt-stressed plants, according to Shakirova et al. [51].

In this experiment, a 4-fold lower GA4 content was observed in mandarin leaves at the beginning of phase III of trees irrigated with RW, while GA3 levels were higher (Table 3). Gibberellins are known to stimulate vegetative growth to achieve high leaf area and photosynthetic rates and are a controlling factor in plant architecture [52]. Berli et al. [53] observed that GA4 applications increased berries per bunch. The low level of GA4 in RW may have contributed to lower stressed plant growth (canopy growth) and fruit growth (expressed as diameter and weight) (Table 4), as mentioned in Section 3.3. On the other hand, the levels of GA3 in the RW may be involved in changing the distribution of Ca^2+^ in the tissue. RW has high Ca^2+^ concentrations. This is not reflected in the leaf mineral composition at phase III of mandarin irrigated with RW. The high GA3 concentration in RW may have limited Ca^2+^ uptake and concentration in the apoplast [54].

The presence of JA was slightly higher in RW. However, this difference was not significant (Tukey, *p* ≤ 0.05) (Table 3). This slight increase could be linked to the root exudate of this phytohormone in citrus fruits [38]. SA and JA can be applied exogenously, and they can control various biochemical and physiological responses together with cellular regulators to mitigate abiotic stressors. In this sense, exogenous SA and JA application treatments alleviate the adverse effect of NaCl on sweet orange growth and chlorophyll content [24].

In the leaves of trees irrigated with RW, a higher ABA content was observed (Table 3). ABA is considered to be the main regulator of plant responses to water deficit or salinity. It promotes growth reduction, leaf senescence, stomatal closure, osmotic adjustment, increased root conductance, and gene induction [55]. ABA under salt stress conditions could be potentiated in terms of its synthesis under RW salt stress destined for exudation at the root level, a behavior previously described in citrus [38]. It should be noted that the ABA-activated signaling pathway is affected as a molecular mechanism of the citrus root response to dehydration and salt treatment [56]. In addition, the higher presence of nitrogen species in RW could be linked to a higher concentration of nitrogen of ammonia origin that induces the defensive response of citrus through the priming of this phytohormone against the same salt stress caused by RW [57]. In addition, this phytohormone is directly linked to ACC. Severe water stress is known to disrupt xylem flow and cause ABA to accumulate in stressed roots. ABA then induces the accumulation of ACC. An inverse relationship between ACC and ABA was observed in the treatments tested. It is possible that the high ABA content in RW at the time of observation is later converted to ACC and then to ethylene. At the same time, the high concentration of SA in the RW may be inhibiting the biosynthesis of ethylene (as mentioned above). Therefore, the delayed ripening of RW fruit observed in the color parameters would be explained by higher concentrations of ABA (a consequence of low ACC concentration) and SA. Ethylene is responsible for the ripening of the fruit. It accelerates the loss of color through the breakdown of chlorophyll and the deposition of carotenoids in ripe citrus fruit [15]. The low concentration of ACC in RW treatment could explain the delayed fruit ripening in plants irrigated with RW.

Finally, the concentrations of Zr and IPA did not differ between the treatments (Tukey, *p* ≤ 0.05) (Table 3).

### 2.3. Response of Agronomic Parameters to Saline Reclaimed Water: Tree Canopy, Yield, and Fruit Quality

Parameters related to growth canopy, yield, and fruit quality varied according to the irrigation water quality used in the different treatments. Compared with the control, trees irrigated with RW showed a significant reduction in growth canopy of 10.3%. These data are consistent with observations by Nicolás et al. [58], who observed an average 18% canopy reduction over the 6-year experiment, and by Romero-Trigueros et al. [13] and Pérez-Pérez et al. [28], who cited that irrigation with saline RW in citrus is generally associated with a reduction in the tree canopy.

Therefore, the maintenance of osmotic stress due to the high levels of all these ions, which results in additional energy consumption, along with the nutritional status of the plants, may be responsible for the lower metabolic activity in plants irrigated with reclaimed water (RW). This deficit can be expressed in a reduced biosynthesis of amino acids, which results in a lower growth rate of the tree, a lower observed tree canopy, and, possibly, a smaller fruit size. Other authors found that more than 11 years of irrigation, with reclaimed waters in the same mandarins, were not sufficient to promote changes in their leaf anatomies [59], but they did find differences in tree size.

Yields were 68.49 and 58.18 kg pl^−1^ for FW and RW (Table 4). Overall, these values were slightly lower than those observed on the same fruit orchard by Romero-Trigueros et al. [3] during the period 2008 to 2015 (96.20 and 79.55 kg pl^−1^) for the FW and RW treatments, respectively. Mandarin tree yield did not differ between treatments, although yield was 15% lower in RW than in FW (Table 4). However, this difference was not significant (Tukey *p* > 0.05), similar to results found in (Pereira et al., 2012) [60], where RW irrigation changed many soil parameters, but no difference in citrus yield was observed. Pérez-Pérez et al. [33] also observed no reduction in growth in a stand of adult “Clemenules” (*Citrus reticulata* Blanco) grafted on “Cleopatra” (*Citrus reshni* Hort. Ex Tanaka), irrigated with saline water (40 mM ClNa) for three seasons. Conversely, Romero-Trigueros et al. [3] observed a significant decrease (17.3%) in total production (kg per tree) in plants irrigated with RW compared with those irrigated with FW (average of the eight growing seasons analyzed, period 2008–2015).

The FN did not change with the treatment. As leaf-derived cytokinins (zeatin) have a role in stimulating bud growth in evergreen trees [19], the high concentration of tZ in the leaf observed in RW may have contributed to the good fruit set achieved in RW.

Significant differences between the treatments were observed for the different quality parameters in mandarin trees. Regarding fruit weight, in this trial RW caused a reduction in weight and diameter of harvested fruit of 13.72 and 5.95%, respectively, compared with FW (Table 4). In previous studies by Romero-Trigueros et al. [3], in the same bush, an increase in fruit weight was observed in the short term (during the period 2008–2011); in the medium term, there was no change (2012–2014), and subsequently a tendency to decrease was observed. Other authors did not find any differences in the fruit weight of Clemenules mandarin trees irrigated with a 40 mM saline solution, on average, over the three years [28].

The peel thickness showed a 48% reduction in fruits from RW irrigation compared with FW. Pérez-Pérez et al. [28] observed an increase in PT in the first year and a decrease in the second year in “Clemenules” fruits irrigated with saline solution (40 mM). Romero-Trigueros et al. [3], evaluating PT over time in trees irrigated with reclaimed water, observed that PT decreased. The high K^+^ content of reclaimed irrigation water may explain this result (Table 1). Embleton and Jones [61] showed that increasing K^+^ (4–11 kg per tree) reduced lemon peel thickness.

With regard to the color of the fruit, during ripening, citrus fruit peel undergoes “color break”, a process characterized by the conversion of chloroplast to chromoplast. The process involves the progressive loss of chlorophylls and the gain of carotenoids, changing peel color from green to orange [62]. In the trial, the Chroma values (72.63 ± 1.92 and 67.85 ± 4.09 in FW and RW, respectively) were significantly lower in RW than in FW. Hue values in RW were significantly higher than in FW (74.09 ± 3.12 and 83.29 ± 4.63 in FW and RW, respectively). This would reflect an immature fruit. The RW treatment gave the mandarin lower color indices at harvest, as observed in Pérez-Pérez et al. [28] under the trial conditions; thus, under saline (these) conditions, it would be necessary to delay the harvest until the fruits reached an optimum color index [28].

The quantity and quality of the fruit juice change according to the irrigation water. RW treatment caused juice content to decrease by 9.9%. Depending on the variety of mandarin and the weather of the treatment, JC was reduced or kept constant in experiments with two varieties of mandarin trees irrigated with 40 mM NaCl [28].

In this study, fruit harvested in RW had a 15.2% decrease in SSC and a 42.4% increase in TA; hence, a 40.7% increase in MI in comparison to the control. This was due to the increase in acidity. This was accompanied by a greener appearance (color) of the fruit. These parameters may show different results: Pérez-Pérez et al. [28] observed an increase in SSC and TA in both mandarins due to the saline treatment. Romero-Trigueros et al. [3] observed a decrease in SSC, but not in TA when treated with RW in the same orchard for four seasons.

### 2.4. Correlation Between the Main Agrometabolic and Agronomic Parameters

Finally, a simplified correlation matrix, taking into account the parameters that showed significant differences between treatments (FW and RW) and also correlated with the highest number of fruit quality parameters analyzed, was carried out to confirm the relationships observed above. Table 5 shows the results.

The values of the Cl^−^ concentration showed a positive and significant correlation with the phytohormones SA and GA3 and a negative correlation with the agronomic parameters W, D, and JC. B and Br^−^ concentrations correlated with each other. Both of them were negatively and significantly correlated with Ala and GA4, as well as with CG and PT, and positively correlated with ABA. Overall, Cl^−^ and Br^−^ ions and B are detrimental to the growth and production of mandarin trees. They induce a stress response by activating the hormones SA, GA3, and ABA. The production of osmolytes is directly related to the positive correlation between phytotoxic ions and phytohormones, especially SA and ABA. These osmolytes enable the plant to adapt to the cellular environment to high osmotic gradients caused by continuous excess ions [63]. In particular, phytohormone SA correlated with 70% of matrix parameters. SA is a common, plant-produced signal molecule that is responsible for inducing tolerance to a number of biotic and abiotic stresses. The adverse effects of salt stress in citrus could be alleviated by exogenous application of SA [23].

The phytohormones SA, GA4, and ABA were correlated with the amino acid Ala. The hormones correlated to one another (ACC and tZ, SA and GA4, SA and ABA, SA and GA3, GA4, and ABA, GA4, and GA3) (Table 5). Changes in cell division and elongation are mediated by several hormones: auxin, cytokinin, gibberellin, and abscisic acid. In citrus rootstocks, smaller rootstocks have been shown to contain lower growth-promoting hormones (GA, IAA, and Ck) and higher growth-inhibiting hormones (ABA) [52].

There was a negative correlation between GC and the concentration of B, Br^−^ (Table 5). These ions exert a phytotoxic effect. This is reflected in a reduction in the growth and productivity of the fruit tree. This is added to the energetic cost of absorbing them [30]. Khan et al. [64] investigated the nutrient uptake of Salustiana sweet oranges grafted on five rootstocks and showed that the size-controlled rootstock range had a significant effect on the uptake capacity of macro- and micronutrients [52].

Furthermore, GC correlated negatively with the phytohormone ABA and positively with Ala and GA4 (Table 5). Alanine, arginine, and glycine are involved in chlorophyll synthesis [22]. Gibberellins are a controlling factor in plant architecture and are known to stimulate vegetative growth to achieve high leaf area and photosynthetic rates. The expansion of the canopy, the chlorophyll content, the stomatal conductance, and the photosynthetic activity are also directly affected by changes in plant growth [52].

Toxic ions correlated with fruit quality: W was negatively correlated with Cl^−^ and B. D and JC were correlated with Cl^−^ (Table 5). It is possible that the effect of the ions in reducing the osmotic potential of the cells is responsible for the higher juice content in RW.

W was positively correlated with GA4 and ACC and negatively correlated with tZ and SA. There was a correlation between fruit and juice quality parameters: D, PT, JC, and TA were positively correlated with W. SSC correlated negatively with TA, and both correlated with MI (Table 5). The combination of quality parameters can lead to changes in taste and flavor, and these can be modified by salinity stress. Depending on the citrus cultivar and the total amount of irrigation water, salinity can have a positive or negative effect on fruit quality [28].

## 3. Materials and Methods

### 3.1. Experimental Plot and Irrigation Treatments

The experiment was performed during 2018 in a commercial 0.5 ha orchard cultivated with mandarin trees (*Citrus Clementina* cv. “Orogrande”) grafted on Carrizo citrange *(Citrus sinensis* [L.] Osb. × *Poncirus trifoliata* [L.]) rootstock. All mandarins were considered adult trees. The orchard was located in Campotéjar Murcia, Spain (38°07′18″ N; 1°13′15″ W), where the climate is Mediterranean semi-arid with warm, dry summers and mild winter conditions. Specific climate records for this experiment were obtained daily from a weather station near experimental orchard.

Soils within the first 90 cm depth had a loamy texture (typical according to Soil Survey Staff, 2014) [65] and bulk density of 1.37 g cm^−3^.

A total of 96 trees, spaced at 3.5 m between plants and 5 m between rows, were used in this study. The experimental design was a randomized complete design with four blocks and four experimental plots per block. The standard plot, which covered about 210 m^2^, was made up of twelve trees, organized in 3 adjacent rows with 4 trees per row. In each plot, the two central trees of the middle row, hereafter called “inner trees”, were used for measurements, and the other ten trees were guard trees.

The samplings were carried out at phase III fruit maturity. Citrus tolerance to salt is at its maximum at this stage. At this stage, the fruit practically stops growing. All the changes associated with ripening take place, and fruits undergo a non-climacteric process [15].

Trees were irrigated to fully meet crop water requirements (100% of crop evapotranspiration), calculated using the Penman–Monteith method [66], daily throughout 2018.

Treatment consisted of two water qualities: fresh (FW) and reclaimed (RW) water. FW was pumped from the “Tajo-Segura” water transfer canal. RW was pumped from a nearby wastewater treatment plant. The RW is a tertiary reclaimed water.

Both treatments received the same amounts of fertilizer (N–P_2_O_5_–K_2_O), applied through the drip irrigation system (215–100–90 kg ha^−1^ year^−1^). Pest control practices and pruning were those commonly used by growers in the area, and no weeds were allowed to develop within the orchard.

### 3.2. Water Analyses

Twelve water samples from each irrigation source were collected monthly in 2018 in glass bottles, transported in an ice chest to the laboratory, and stored at 5 °C before being processed for physical and chemical analyses. An inductively coupled plasma mass spectrometer (ICP-ICAP 6500 DUO Thermo, Manchester, UK) was used to determine the concentration of Ca^2+^, Mg^2+^, K^+^, Na^+^, and total B. Anions (Cl^−^, NO_3_^−^, PO_4_^3−^, and SO_4_^2−^) were analyzed by ion chromatography with a liquid chromatograph (Metrohm, Herisau, Switzerland). Electrical conductivity of water (EC_w_) was determined using a PC-2700 m (Eutech Instruments, Singapore), and pH was measured with a Crison 507 pH meter (Crison Instruments S.A., Barcelona, Spain).

### 3.3. Chemicals

Cation exchange resin, Dowex (H^+^) 50WX8-400, acetonitrile, methanol, *o*-phthalaldehyde (OPA)/2 mercaptoethanol, disodium hydrogen phosphate dihydrate, propionic acid, standards of amino acids (L-alanine (Ala), L-arginine (Arg), L-asparagine (Asn), L-aspartic acid (Asp), glycine (Gly), L-glutamic acid (Glu), L-glutamine (Gln), L-histidine (His), L-isoleucine (Ile), leucine (Leu), L-methionine (Met), L-phenylalanine (Phe), L-serine (Ser), L-threonine (Thr), L-tryptophan (Trp), L-tyrosine (Tyr), and L-valine (Val)) were obtained from Sigma-Aldrich (Madrid, Spain). Phytohormones (1-aminocyclopropane-l-carboxylic acid (ACC), trans-zeatin (tZ), zeatin riboside (Zr), isopentenil adenine (IPA), gibberelic acid GA1, GA3, GA4 were obtained from Olchem (Olomouc, Czech Republic), and indoleacetic acid (AIA), abscisic acid (ABA), salicylic acid (SA), jasmonic acid (JA)) from Santa Cruz Biotechnologies (Dallas, TX, USA). The solid-phase extraction (SPE) cartridges used with clean-up of sample purposes were Strata cartridges (Strata X-AW, 100 mg/3 mL), which were acquired from Phenomenex (Torrance, CA, USA).

### 3.4. Nutrient and Agrometabolites at Leaf Level

#### 3.4.1. Nutritional Status

To perform the analyses, at the beginning of the third phase, twenty leaves per tree from non-fruiting branches in the central part of the tree were collected. Leaves were collected early in the morning. They were transported to the laboratory in refrigerated plastic bags. An inductively coupled plasma mass spectrometer (ICP-ICAP 6500 DUO Thermo, UK) was used to determine the total concentration of total B, Na^+^, K^+^, Ca^+2^, and Mg^2+^. Anions (Cl^−^, NO_3_^−^, PO_4_^3−^, and SO_4_^2−^) were analyzed by ion chromatography with a liquid chromatograph (Metrohm, Herisau, Switzerland). Total N was determined following the Dumas method [67].

#### 3.4.2. Free Amino Acid Extraction and Chromatographic Determination

The extraction method for free amino acids was according to Collado-González et al. [68] and Cerrillo et al. [69]. Briefly, fresh leaves were crushed in a mortar with liquid nitrogen and homogenized with methanol (MeOH)/water, 1:1, *v*/*v* using an ultraturrax (IKA, T10, Staufen, Germany). The homogenates were sonicated and centrifuged. The extracts were immediately derivatized. The derivatization of amino acids was accomplished by following the Waters AccQTagTM Ultra UHPLC amino analysis procedures, as described by Gómez-Bellot et al. [6], Collado-González et al. [68], and Cerrillo et al. [69].

These compounds were identified using an ultra-high performance liquid chromatography (UHPLC) system coupled to a 6460 tandem mass spectrometer (UHPLC-ESI-QqQ-MS/MS) (Agilent Technologies, Waldbronn, Germany). Data acquisition and processing were performed using MassHunter software version B.04.00 from Agilent Technologies. Mobile phase A consisted of 50 mL of an aqueous solution (acetonitrile, formic acid, and 5 mM ammonium acetate in water) (10:6:84, *v*/*v*/*v*) diluted with 950 mL of Milli-Q water. Mobile phase B was a mixture of acetonitrile and formic acid (99.9:0.1, *v*/*v*). The injection volume was 20 μL, and the elution was performed at a flow rate of 0.5 mL/min^−1^. The gradient profile was 99.9%A at 0–0.5 min, 90.9%A at 5.7 min, 78.8%A at 7.7 min, 40.4%A at 8–10 min, 10%A at 10.01–12.00 min, and 99.9%A at 12.01–14.00 min. The mass spectrometry (MS) analysis was applied in the multiple reaction monitoring (MRM) mode, which was performed using the positive ionization mode. The working conditions for the MS parameters of the electrospray source were as follows: gas flow, 9 L min^−1^; nebulizer, 40 psi; capillary voltage, 4000 V; nozzle voltage, 1000 V; gas temperature, 325 °C; sheath gas temperature, 390 °C; and jet stream gas flow, 11 L min^−1^. The acquisition time was 12 min for each sample. The most abundant MRM transition of each analyte was used for amino acid quantitation by comparison with its corresponding authentic standard. Calibration standard curves were generated using individual amino acids prepared by dissolving each amine in Bis–Tris (pH 6.5) [68].

#### 3.4.3. Qualitative and Quantitative Determination of Phytohormones

The qualitative and quantitative determination of phytohormones was carried out according to Gómez-Bellot et al. [6].

Briefly, samples of powder (0.1 g) were pestled with 0.5 mL 80% methanol/water (*v*/*v*). The extracts were centrifuged at 20,627× *g* for 15 min at 4 °C, and the supernatants were passed through Chromafix C_18_ solid-phase extraction cartridge (Macherey Nagel, Düren, Germany). The eluted sample was concentrated to dryness (Speedvac, Thermo, Waltham, MA, USA). Then, the dry residue was resuspended with 200 µL of 20% methanol/water (*v*/*v*), sonicated for 8 min, and filtered through 0.45 µm polyethersulfone filter (Millipore, Madrid, Spain) and finally injected into an ultra-high-performance liquid chromatography (UHPLC) coupled triple quadrupole mass spectrometry (UHPLC-ESI-QqQ-MS/MS) for qualitative and quantitative analysis. Chromatographic separation of phytohormones was performed using a UHPLC coupled to a 6460 triple quadrupole-MS/MS (Agilent Technologies, Waldbronn, Germany), using a BEH C_18_ analytical column (2.1 × 50 mm, 1.7 µm) (Waters, Milford, MA, USA) applying the chromatographic, ionization, and fragmentation conditions described by Gómez-Bellot et al. [6]. Data acquisition and processing were performed using Mass Hunter software version B.08.00 (Agilent Technologies, Waldbronn, Germany). The concentration of the phytohormones detected in the analyzed leaf sample was quantified by calculating the area under the curve, taking as reference the standard curves of known concentrations just prepared each day of analysis.

### 3.5. Agronomic Parameters: Growth Canopy, Yield, and Quality Fruit

The growth canopy was estimated from tree height and perimeter of all trees with a pole marked at every half meter in two perpendicular directions. The formula proposed by Hutchinson [70] was employed, which considers the tree as pyramid-shaped unit.

Eight inner trees per treatment were evaluated to determine (NF) number of fruits per tree, (Y) yield in total kilograms per tree, (W) fruit weight, and (D) fruit diameters. The fruits were harvested on the 13 November. Fruit quality assessment was by random selection of 100 fruits per treatment (25 fruits per block). The parameters evaluated included the peel thickness (PT), color, juice content (JC), soluble solid content (SSC), titratable acidity (TA), and maturity index (MI). A digital caliper was used to measure the thickness of the skin. The color was measured in two areas of the equatorial plane of the fruit using a Minolta CR-300 color meter. The color peel was calculated using Chroma (chroma = (a*2 + b*2)1/2), which is the perpendicular distance from the brightness axis, indicating the color intensity and hue (Hue = tan^−1^(b*/a*)), which is the angle formed by a line from the origin to the intercept of a* (*x*-axis) and b* (*y*-axis) coordinates, where 0° = red, 90° = yellow, 180° = green, and 270° = blue, and indicates the visual property normally termed color [71]. JC was calculated as juice weight/fruit weight × 100. In order to determine SSC and TA, fifty milliliters of juice per fruit were used. SSC was determined with a handheld refractometer (Atago N1, Tokyo, Japan) and TA by titrating 10 mL of juice with 0.1 mol L^−1^ NaOH to pH 8.1 using an automatic titration system. Finally, MI, which affects the perception of taste (sweetness and acidity) by the consumer, was computed as the ratio of SSC to TA. In this case, the SSC/TA ratio indicated both fruit maturity in the field and postharvest consumer perception of juice quality.

### 3.6. Statistical Analysis

All assays were developed in triplicate (*n* = 4), and the values were provided as means ± the standard error of the mean (SD). Statistical analysis was performed as a weighted analysis of variance (ANOVA; statistical software Statgraphics Centurion XVI (version 16) (StatPoint Technologies Inc., Warrenton, VA, USA). The differences between means were compared by the multiple range test of Tukey, and the degree of significance was setup at *p* < 0.05. A Pearson correlation analysis was performed. The parameters that showed significant differences between treatments (FW and RW) were taken into account. Depending on the positive or negative correlation between pairs of variables, Pearson’s coefficient has values between +1 and −1.

## 4. Conclusions

Citrus trees are very sensitive to salinity conditions, so this is a good result considering the long period of time the plants were tested and the significant reduction in fruit yield over a long period of exposure to salinity. This graft/rootstock combination kept Na concentrations relatively stable compared with control leaves. The twofold lower concentration of amino acids in RW suggests a lower nutrient level due to salt stress and consequently a lower metabolic capacity for protein synthesis and ultimately lower growth. Irrigation with saline water also resulted in different vegetative growth patterns. This may be related to salinity affecting photoassimilate partitioning in trees, prioritizing vegetative development (reflected in fruit number) over canopy vegetative growth. Fertilization with serine, alanine, or glycine would help to increase plant growth. In general, changes in hormone levels lead to a reduction in growth and an increase in stress tolerance. The opposing effects of phytohormones on growth and stress tolerance can be paralleled. Apparently, stress tolerance is incompatible with optimal growth. This antagonism is coordinated by different hormone signaling pathways. The hormonal balance at the beginning of fruit ripening shows a deficit of ACC and GA4. This may be involved in the delay of fruit ripening in plants under salt stress. In these cases, it is recommended to delay the harvesting of plants under salt stress. This will allow the fruit to ripen. Finally, based on the results obtained, it could be assumed that exogenous applications of L-alanine could optimize vegetative growth, given the high correlation of the amino acid Ala with GC. This would allow a higher production of photoassimilates, which in turn could be used in phase II of fruit growth for the increase in W and D. In addition, SA has a strong effect on the parameters evaluated under salt stress, both in the present experiment and in the literature consulted. SA applications could increase the net photosynthetic rate and contribute to the best ameliorative remedies by helping to improve plant tolerance to salt stress. These results show significant advances in the physiological actions that can be exerted by irrigation with reclaimed water; however, further advances are needed to endorse these results in the sense of understanding the mechanisms of the plant in this agronomic environment, with the help of keyomics technologies such as metabolomics, transcriptomics, and proteomics.

## Figures and Tables

**Figure 1 ijms-26-03450-f001:**
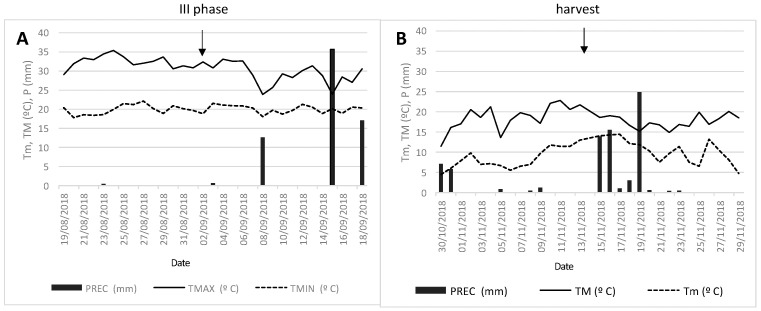
Meteorological records 15 days before and 15 days after the observation date in leaf at the beginning of the third rapid fruit-growing phase (**A**) and at harvest (**B**).

**Table 1 ijms-26-03450-t001:** Physical and chemical analyses of both fresh transfer water (FW) and reclaimed water (RW) for 2018.

Parameter	Unit	FW	RW
EC_w_	dS m	1.38 ± 0.12	3.41 ± 0.13
SAR_w_	(meq L^−1^)^0.5^	3.35 ± 0.69	6.09 ± 1.89
pH		8.10 ± 0.03	7.73 ± 0.06
Ca^2+^	meq L^−1^	4.68 ± 0.70	7.35 ± 0.83
Mg^2+^	meq L^−1^	4.50 ± 5.02	8.27 ± 6.89
K^+^	mg L^−1^	15.76 ± 4.16	31.46 ± 3.37
Na^+^	meq L^−1^	7.19 ± 3.20	17.03 ± 4.10
B	mg L^−1^	0.19 ± 0.04	0.60 ± 0.08
Cl^−^	meq L^−^	6.35 ± 3.98	17.10 ± 3.63
NO_3_^−^	mg L^−1^	3.89 ± 0.73	10.50 ± 2.27
PO_4_^3−^	mg L^−1^	1.14 ± 1.58	2.86 ± 0.87
SO_4_^2−^	meq L^−1^	7.92 ± 5.10	14.90 ± 6.20

Values are averages ± SE (standard error) of 12 individual samples collected during the experimental period. EC_w_: electrical conductivity (dS m^−1^); SAR_w_: sodium adsorption ratio; FW: fresh transfer water; RW: reclaimed water.

**Table 2 ijms-26-03450-t002:** Physical and chemical analyses of mandarin leaf at phase III of maturity fruit for both fresh transfer water (FW) and reclaimed water (RW).

Toxicity Elements				Macro Nutrients				Micronutrients			
	Units	FW	RW		Units	FW	RW		Units	FW	RW
Na^+^	% dw	0.05 ± 0.03 a	0.05 ± 0.01 a	N total	% dw	2.09 ± 0.08 a	1.91 ± 0.07 a	Cu	mg kg^−1^	4.55 ± 0.68 a	4.74 ± 0.61 a
Cl^−^	% dw	0.40 ± 0.13 a	0.98 ± 0.12 b	P	% dw	0.09 ± 0 a	0.08 ± 0 a	Zn^2+^	mg kg^−1^	12.7 ± 0.83 a	11.14 ± 0.55 a
B	mg kg^−1^	83.64 ± 11.54 a	169.91 ± 6.70 b	K^+^	% dw	1.01 ± 0.11 a	0.88 ± 0.03 a	Fe	mg kg^−1^	138.27 ± 5.67 a	97.47 ± 17.12 a
Li^+^	mg kg^−1^	6.81 ± 0.44 a	10.48 ± 0.83 b	Ca^2+^	% dw	4.07 ± 0.06 a	3.08 ± 0.11 a	Mn	mg kg^−1^	38.54 ± 1.31 a	37.61 ± 1.22 a
Br^−^	% dw	0.004 ± 0 a	0.01 ± 0 b	Mg^2+^	% dw	0.39 ± 0.02 a	0.38 ± 0.01 a	Ni^3^	mg kg^−1^	0.93 ± 0.07 a	1.09 ± 0.07 a
				S	% dw	0.24 ± 0.01 b	0.21 ± 0.01 a				
				NO_3_^−^	% dw	0.34 ± 0.43 a	0.84 ± 0.25 a				
				PO_4_^3−^	% dw	1.62 ± 0.15 a	2.23 ± 0.61 a				
				SO_4_^2−^	% dw	1.48 ± 0.45 a	2.27 ± 0.09 a				

Toxicity elements: Na^+^, Cl^−^, B, Li^+^, and Br^−^; macronutrient: N, P, S, NO_3_^−^, PO_4_^3−^, SO_4_^2−^, Ca^2+^, Mg^2+^, and K^+^; and micronutrients: Cu, Zn^2+^, Fe, Mn, and Ni. Data presented as mean ± standard deviation (SD). Values in the same row followed by different letters are significantly different at *p* < 0.05 according to one-way analysis of variance (ANOVA) and multiple range tests of Tukey.

**Table 3 ijms-26-03450-t003:** Amino acid and phytohormone analyses of mandarin leaf at phase III of maturity fruit for both fresh transfer water (FW) and reclaimed water (RW).

	Amino Acids (ng g^−1^) *			Phytohormones (ng g^−1^) *	
	FW	RW		FW	RW
Ala	32,882.50 ± 79.43 b	10,915.80 ± 1318.94 a	ACC	2563.85 ± 344.29 b	1664.95 ± 78.98 a
Pro	10,994.40 ± 869.72 a	6403.35 ± 2027.23 a	*tZ*	709.377 ± 126.13 a	1150.7 ± 43.23 b
Tyr	2450.23 ± 1010.53 a	2597.47 ± 237.78 a	SA	127.66 ± 17.08 a	327.55 ± 44.28 b
GABA	1411.76 ± 332.792 a	816.86 ± 108.39 a	GA4	95.64 ± 12.62 b	20.59 ± 4.56 a
Ser	1119.24 ± 73.54 b	607.38 ± 182.69 a	JA	4.95 ± 0.60 a	5.29 ± 1.13 a
Cys–Gly	878.17 ± 287.304 a	693.27 ± 394.28 a	ABA	0.08 ± 0.05 a	2.54 ± 0.25 b
Leu	95.79 ± 35.23 a	654.13 ± 575.74 a	Zr	1.35 ± 1.62 a	0 ± 0 a
Arg	422.40 ± 60.68 a	321.83 ± 39.09 a	IPA	0.14 ± 0.01 a	0.15 ± 0.01 a
Asn	242.63 ± 63.50 a	267.72 ± 108.00 a	GA3	0 ± 0 a	0.14 ± 0.03 b
Glu	251.53 ± 95.95 a	136.36 ± 30.72 a			
Val	150.80 ± 63.34 a	150.05 ± 25.31 a			
Met	128.61 ± 18.507 a	79.15 ± 25.21 a			
Asp	112.07 ± 3.15 a	50.38 ± 40.17 a			
Gly	89.47 ± 5.46 b	39.59 ± 17.34 a			
Phe	38.80 ± 25.097 a	38.17 ± 27.72 a			
Citrulline	46.40 ± 25.423 a	27.35 ± 4.46 a			
Thr	25.22 ± 4.51 a	22.29 ± 5.00 a			
His	19.45 ± 8.87 a	12.17 ± 3.45 a			

Phytohormones: ACC, 1-aminocyclopropane-l-carboxylic acid; tZ, trans-zeatin; SA, salicylic acid; GA4 and GA3, gibberelic acid; JA, Jasmonic acid; ABA, abscisic acid; Zr, zeatin riboside; IPA, isopentenil adenin. Amino acid: Ala, alanine; Por, proline; Tyr, tyrosine; GABA, G-aminobutiric acid; Ser, serine; Cys–Gly, cysteine–glycine; Leu, leucine; Arg, arginine; Asn, asparagine; Glu, glutamic acid; Val, valine; Met, methionine; Asp, aspartic acid; Gly, glycine; Phe, phenylalanine; Thr, threonine; His, histidine. Data presented as mean ± standard deviation (SD). * Values in the same row followed by different letters are significantly different at *p* < 0.05 according to one-way analysis of variance (ANOVA) and multiple range test of Tukey. In the leaves of mandarins irrigated with FW, the total amino acid content was twice as high as the concentration in RW, with values of 51.3 and 23.83 µg g^−1^, respectively (Table 3). These findings agree with Di Martino et al. [39], who found that amino acid content in last fully expanded leaves decreases to 200% of controls in salt-stressed leaves [35].

**Table 4 ijms-26-03450-t004:** Mean canopy growth, yield, and quality parameter values in mandarin crop for both treatments.

	Units	FW	RW
CG	m^3^	20.26 ± 1.79 b	18.18 ± 2.60 a
Yield	kg pl^−1^	68.49 ± 12.48 a	58.18 ± 13.53 a
FN		809.77 ± 134.09 a	720.88 ± 167.71 a
W	g	93.53 ± 16.13 b	80.70 ± 5.68 a
D	mm	60.04 ± 4.44 b	56.47 ± 1.97 a
PT	mm	3.02 ± 0.40 b	1.57 ± 0.23 a
JC	mL	42.93 ± 4.88 b	38.67 ± 5.77 a
SSC	°Brix	13.77 ± 0.48 b	11.68 ± 0.29 a
TA	%	0.99 ± 0.08 a	1.41 ± 0.11 b
MI		14.04 ± 1.32 b	8.32 ± 0.72 a

Data presented as mean ± SD. Values in the same row followed by different letters are significantly different at *p* < 0.05 according to one-way analysis of variance (ANOVA) and multiple range test of Tukey. Fresh transfer water (FW), reclaimed water (RW), canopy growth (CG), yield; fruit number (FN); fruit weight (W); fruit diameter (D); peel thickness (PT); juice content (JC); soluble solid content (SSC); treatable acid (TA); maturity index (MI).

**Table 5 ijms-26-03450-t005:** Pearson’s correlation matrix between minerals and fruit quality parameters. Pearson correlation coefficient values.

	B	Br^−^	Ala	ACC	tZ	SA	GA4	ABA	GA3	GT	W	D	PT	JC	SSC	TA	MI
Cl^−^	0.9268	0.8850	−0.8955	−0.9195	0.9321	0.9601 *	−0.9368	0.9355	0.9582 *	−0.8183	−0.9854 *	−0.9993 **	−0.9476	−0.9979 **	−0.8573	0.9392	−0.9217
B		0.9893 *	−0.9907 **	−0.9320	0.9687 *	0.9464	−0.9811 *	0.9713 *	0.9366	−0.9680 *	−0.9688 *	−0.9120	−0.9957 **	−0.9218	−0.8941	0.8660	−0.9100
Br^−^			−0.9997 **	−0.8702	0.9231	0.9455	−0.9856 *	0.9789 *	0.9365	−0.9918 **	−0.9286	−0.8673	−0.9872 *	−0.8718	−0.9304	0.8667	−0.9251
Ala				0.8742	−0.9261	−0.9529 *	0.9893 *	−0.9832 *	−0.9444	0.9884 *	0.9355	0.8786	0.9904 **	0.8823	0.9348	−0.8777	0.9325
ACC					−0.9927 **	−0.8369	0.8660	−0.8470	−0.8241	0.8259	0.9655 *	0.9111	0.9161	0.9372	0.6987	−0.7477	0.7606
tZ						0.8825	−0.9157	0.8991	0.8702	−0.8870	−0.9793 *	−0.9211	−0.9554 *	−0.9427	−0.7712	0.7930	−0.8178
SA							−0.9869 *	0.9915 **	0.9996 **	−0.9019	−0.9484	−0.9535 *	−0.9723 *	−0.9416	−0.9671 *	0.9808 *	−0.9917 **
GA4								−0.9990 **	−0.9822 *	0.9588 *	0.9524 *	0.9247	0.9934 **	0.9208	0.9618 *	−0.9378	0.9723 *
ABA									0.9880 *	−0.9504 *	−0.9444	−0.9242	−0.9876 *	−0.9171	−0.9720 *	0.9503 *	−0.9817 *
GA3										−0.8911	−0.9417	−0.9524 *	−0.9651 *	−0.9386	−0.9684 *	0.9860 *	−0.9938 **
GT											0.8773	0.7965	0.9589 *	0.8031	0.9097	−0.8088	0.8860
W												0.9792 *	0.9745 *	0.9875 *	0.8467	−0.8948	0.8999
D													0.9351	0.9975 **	0.8464	−0.9388	0.9152
PT														0.9388	0.9241	−0.9087	0.9433
JC															0.8249	−0.9152	0.8954
SSC																−0.9535 *	0.9877 *
TA																	−0.9869 *

Cl^−^, chloride; B, boron; Br^−^, bromide; Ala, alanine; ACC, 1-aminocyclopropane-l-carboxylic acid; tZ, trans zeatin; SA, salicylic acid; GA4–GA3, gibberelins; ABA, abscisic acid; GC, growth canopy; W, fruit weight; D, fruit diameter; PT, peel thickness; JC, juice content; SSC, soluble solid content; TA, tritatable acid; MI, maturity index. Statistical significance: * = *p* ≤ 0.05; ** = *p* ≤ 0.01.

## Data Availability

The data presented in this study are available on request from the corresponding authors.

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
