# Peer review of "Agronomic and Metabolic Responses of *Citrus clementina* to Long-Term Irrigation with Saline Reclaimed Water as Abiotic Factor"

_ijms, 2025, doi:10.3390/ijms26073450_

Round 1

Reviewer 1 Report

Comments and Suggestions for Authors

After reading this manuscript, I believe it is not suitable for publication in IJMS. Instead, it would be more appropriate for publication in Agronomy or Horticulturae. This is because this manuscript has no connection with molecular biology. Instead, it focuses on the statistics and interpretation of agronomic traits and physiological indicators. Moreover, the data in Table 5 could be completely replaced by figures, as the table is not intuitive and difficult to read. 

Author Response

R1

After reading this manuscript, I believe it is not suitable for publication in IJMS. Instead, it would be more appropriate for publication in Agronomy or Horticulturae. This is because this manuscript has no connection with molecular biology. Instead, it focuses on the statistics and interpretation of agronomic traits and physiological indicators. Moreover, the data in Table 5 could be completely replaced by figures, as the table is not intuitive and difficult to read. 

Response:

The current manuscript is a research study that combines an agronomic and a molecular approach, being the participation of plant hormones and their direct link with the amino acid physiology of the plant a key factor. Therefore, the study carried out is agromolecular as a whole, which is why we consider that it falls within the scope of IJMS. A double vision that allows us to identify molecularly how the plant behaves in the face of irrigation treatments with saline reclaimed water with respect to fresh water directly related to the supply of different minerals available to be absorbed by the plant.

Table 5 representing the Pearson's correlation matrix between minerals and fruit quality parameters is the typical way to outline the different interrelationships with a direct view. A figure could mask the accuracy and precision provided in each of the correlation coefficients, most of which vary from the third or fourth decimal place.

Reviewer 2 Report

Comments and Suggestions for Authors

Authors studied the physiological response of mandarin trees to long period of irrigation with saline reclaimed water (RW) and fresh water (FW). Authors studied the leaf mineral constitution, free amino acids and phytohormones balance, and their influence on yield and fruit quality.

Study is generally well planned and performed, results are significant and interesting to researchers in the field. Figure is of good quality. Obtained results support conclusions.

The weaker part of the study  is the discussion and conclusions part which should be more developed. Also presenting of chemical ions formulas should be more precise. Following comments should be addressed:

1.Line 111; use abbreviations as a standard required in ijms : Acronyms/Abbreviations/Initialisms should be defined the first time they appear in each of three sections: the abstract; the main text; the first figure or table. When defined for the first time, the acronym/abbreviation/initialism should be added in parentheses after the written-out form.

2.Line 115; what is Cis-Gly? The same for Table 3 and decription of Table 3. The Gly is also present in Table 3 what is Cis-Gly?

3.Line 120; Remove word Objective, just show in this sentence objectives of research.

4.Line 112-113; presents Ca, Mg, Cu, B, Fe etc as ions (Ca2+, Mg2+, Cu2+, state if there was a BO33- or other boron ion, etc).

5.Line 233 should be phytohormones

6.Table 1 and 5; check the chemical formulas of ions; for example should be Mg2+, NO3- etc.

7.Line 299-300 write of ions as in table 1. The same for description of Table 2.

  1. Description of table 5; use names chloride, bromide, boron.
  2. If boron ions are presented in the study, state if there was a BO33- or other ion. Check the whole text.

10.The discussion section should be more developed, use for ex ample following or related studies.

Effects of reclaimed waters on spectral properties and leaf traits of citrus orchards.

Contreras S, Pérez-Cutillas P, Santoni CS, Romero-Trigueros C, Pedrero F, Alarcón JJ.Water Environ Res. 2014 Nov;86(11):2242-50. doi: 10.2175/106143014x14062131178637.PMID: 25509529

Nutrients and nonessential elements in soil after 11 years of wastewater irrigation.

Pereira BF, He Z, Stoffella PJ, Montes CR, Melfi AJ, Baligar VC.J Environ Qual. 2012 May-Jun;41(3):920-7. doi: 10.2134/jeq2011.0047.PMID: 22565273

Ammonium enhances resistance to salinity stress in citrus plants.

Fernández-Crespo E, Camañes G, García-Agustín P.J Plant Physiol. 2012 Aug 15;169(12):1183-91. doi: 10.1016/j.jplph.2012.04.011. Epub 2012 Jun 19.PMID: 22721954 

Polyamines reprogram oxidative and nitrosative status and the proteome of citrus plants exposed to salinity stress.

Tanou G, Ziogas V, Belghazi M, Christou A, Filippou P, Job D, Fotopoulos V, Molassiotis A.Plant Cell Environ. 2014 Apr;37(4):864-85. doi: 10.1111/pce.12204. Epub 2013 Oct 31.PMID: 24112028 

Combined analysis of mRNA and miRNA identifies dehydration and salinity responsive key molecular players in citrus roots.

Xie R, Zhang J, Ma Y, Pan X, Dong C, Pang S, He S, Deng L, Yi S, Zheng Y, Lv Q.Sci Rep. 2017 Feb 6;7:42094. doi: 10.1038/srep42094.

Citrus plants exude proline and phytohormones under abiotic stress conditions.

Vives-Peris V, Gómez-Cadenas A, Pérez-Clemente RM.Plant Cell Rep. 2017 Dec;36(12):1971-1984. doi: 10.1007/s00299-017-2214-0. Epub 2017 Oct 16.

The Application of a Commercially Available Citrus-Based Extract Mitigates Moderate NaCl-Stress in Arabidopsis thaliana Plants.

Loubser J, Hills P.Plants (Basel). 2020 Aug 10;9(8):1010. doi: 10.3390/plants9081010.

Morphological, physiological, and molecular scion traits are determinant for salt-stress tolerance of grafted citrus plants.

Vives-Peris V, López-Climent MF, Moliner-Sabater M, Gómez-Cadenas A, Pérez-Clemente RM.Front Plant Sci. 2023 Apr 20;14:1145625. doi: 10.3389/fpls.2023.1145625. eCollection 2023.

Soil temperature, nitrogen concentration, and residence time affect nitrogen uptake efficiency in citrus.

Scholberg JM, Parsons LR, Wheaton TA, McNeal BL, Morgan KT.J Environ Qual. 2002 May-Jun;31(3):759-68. doi: 10.2134/jeq2002.7590.PMID: 12026079

  1. Conclusions

Authors should pesent the future directions of research, for example transcriptomic or proteomic studies to describe mechanisms of plant reaction to reused water.

Author Response

Authors studied the physiological response of mandarin trees to long period of irrigation with saline reclaimed water (RW) and fresh water (FW). Authors studied the leaf mineral constitution, free amino acids and phytohormones balance, and their influence on yield and fruit quality.

Study is generally well planned and performed, results are significant and interesting to researchers in the field. Figure is of good quality. Obtained results support conclusions.

The weaker part of the study is the discussion and conclusions part which should be more developed. Also presenting of chemical ions formulas should be more precise. Following comments should be addressed:

1.Line 111; use abbreviations as a standard required in ijms Acronyms/Abbreviations/Initialisms should be defined the first time they appear in each of three sections: the abstract; the main text; the first figure or table. When defined for the first time, the acronym/abbreviation/initialism should be added in parentheses after the written-out form.

Response:

Thank you very much for the comment. In our manuscript we have described the abbreviations in a general section, as we have observed in other IJMS publications. However, in each section we have also detailed each of the abbreviations in parentheses. We have reviewed the entire text and noted that we have overlooked the explanation of some abbreviations, so we have carried out this task in lines 193, 198-200, 207 and in Tables 2 and 3.

2.Line 115; what is Cis-Gly? The same for Table 3 and decription of Table 3. The Gly is also present in Table 3 what is Cis-Gly?

Response:

Thank you for pointing this out. Cis-Gly is a combination by covalent bonding of cysteine and glycine. It was not well explained either in the abbreviations section or in the text. In addition, we have changed the abbreviation Cis-Gly to Cys-Gly as it is more appropriate and accurate.3.Line 120; Remove word Objective, just show in this sentence objectives of research.

4.Line 112-113; presents Ca, Mg, Cu, B, Fe etc as ions (Ca2+, Mg2+, Cu2+, state if there was a BO33- or other boron ion, etc).

Response:

Boron is not usually found as B+3. In fact, it is usually found combined with other elements to form the borate anion.

The measurement method used in this work is the ICP method, and this device measures total boron. Since it is not possible to determine the element's energy state using this method (because there are several options), it cannot be indicated as a catión or anion, but rather as a total element.

Therefore, it is the total B.

The rest of elements occurr some similar. When it is sure that the element is a cation and only have one valency, we have indicated the exact form (cation). When the element have two or more valency, we have indicated total element because the measurement method does not distinguish between valencies.

5.Line 233 should be phytohormones

Reponse:

We agree with this comment. Thanks. The word has been corrected.

6.Table 1 and 5; check the chemical formulas of ions; for example should be Mg2+, NO3- etc.

Response:

Thank you for pointing this out. Ions have been written according to the reviewer's indications.

7.Line 299-300 write of ions as in table 1. The same for description of Table 2.

Response:

Ions have been written according to the reviewer's indications.

  1. Description of table 5; use names chloride, bromide, boron.

Response:

We agree with this comment. The changes have been made.

  1. If boron ions are presented in the study, state if there was a BO33-or other ion. Check the whole text.

Response:

Changes have been made throughout the text according to the response of the question number 4.

10.The discussion section should be more developed, use for ex ample following or related studies.

Response:

Thank you for pointing this out. The discussion section has been more developed using several related studies.

 We have added and disscussed about the next studies:

  1. Scholberg JM, Parsons LR, Wheaton TA, McNeal BL, Morgan KT.J Environ Qual. 2002 May-Jun;31(3):759-68. doi: 10.2134/jeq2002.7590.PMID: 12026079 Soil temperature, nitrogen concentration, and residence time affect nitrogen uptake efficiency in citrus.2. Contreras S, Pérez-Cutillas P, Santoni CS, Romero-Trigueros C, Pedrero F, Alarcón JJ.Water Environ Res. 2014 Nov;86(11):2242-50. doi: 10.2175/106143014x14062131178637.PMID: 25509529 Effects of reclaimedwaters on spectral properties and leaf traits of citrusorchards.
  2. Pereira BF, He Z, Stoffella PJ, Montes CR, Melfi AJ, Baligar VC.J Environ Qual. 2012 May-Jun;41(3):920-7. doi: 10.2134/jeq2011.0047.PMID: 22565273. Nutrients and nonessential elements in soil after 11 years of wastewater irrigation.
  3. Romero-Trigueros, C.; Nortes, P.A.; Alarcón, J.J.; Nicolás, E. Determination of 15N stable isotope natural abundances for assessing the use of saline reclaimed water in grapefruit. Environ. Eng. Manag. J. 2014, 13, 2525–2530 (This reference has been added in the Introduction section)
  4. Xie R, Zhang J, Ma Y, Pan X, Dong C, Pang S, He S, Deng L, Yi S, Zheng Y, Lv Q.Sci Rep. 2017 Feb 6;7:42094. doi: 10.1038/srep42094. Combined analysis of mRNA and miRNA identifies dehydration and salinityresponsive key molecular players in citrusroots.
  5. Vives-Peris V, Gómez-Cadenas A, Pérez-Clemente RM.Plant Cell Rep. 2017 Dec;36(12):1971-1984. doi: 10.1007/s00299-017-2214-0. Epub 2017 Oct 16. Citrusplants exude proline and phytohormones under abiotic stress conditions.
  6. Fernández-Crespo E, Camañes G, García-Agustín P.J Plant Physiol. 2012 Aug 15;169(12):1183-91. doi: 10.1016/j.jplph.2012.04.011. Epub 2012 Jun 19.PMID: 22721954.Ammonium enhances resistance to salinitystress in citrusplants.
  7. Vives-Peris V, López-Climent MF, Moliner-Sabater M, Gómez-Cadenas A, Pérez-Clemente RM.Front Plant Sci. 2023 Apr 20;14:1145625. doi: 10.3389/fpls.2023.1145625. eCollection 2023. Morphological, physiological, and molecular scion traits are determinant for salt-stress tolerance of grafted citrus plants.

The following two references have not been included in the work because:

  • the first refers to the polyamines putrescine, spermidine, and spermine, which, it is true, could have been generated in the reclaimed water due to its previous load with organic matter, but which were not actually analyzed in our study, and it would be too much to speculate about something that could be present but which we have not described.

 the second has as its object of study in one of them citrus extracts and we did not evaluate this issue in our work.1.Tanou G, Ziogas V, Belghazi M, Christou A, Filippou P, Job D, Fotopoulos V, Molassiotis A.Plant Cell Environ. 2014 Apr;37(4):864-85. doi: 10.1111/pce.12204. Epub 2013 Oct 31.PMID: 24112028  Polyamines reprogram oxidative and nitrosative status and the proteome of citrus plants exposed to salinity stress.

2.Loubser J, Hills P.Plants (Basel). 2020 Aug 10;9(8):1010. doi: 10.3390/plants9081010.The Application of a Commercially Available Citrus-Based Extract Mitigates Moderate NaCl-Stress in Arabidopsis thaliana Plants.

  1. Conclusions

Authors should pesent the future directions of research, for example transcriptomic or proteomic studies to describe mechanisms of plant reaction to reused water.

Response:

We agree with this comment. We have added a paragraph to this effect at the end of the conclusions section. (Lines 598-602)

Reviewer 3 Report

Comments and Suggestions for Authors

In this manuscript, David Auñón-Calles and colleagues examined the physiological response of mandarin trees in a semi-arid area effects over a long period of irrigation with saline reclaimed water (RW) and fresh water (FW) in terms of leaf mineral constitution, free amino acids and phytohormones balance, and their influence on yield and fruit quality. I have following comments:

1, For the title, I suggest to employ “Agronomic And Metabolic Responses Of Citrus clementina to Long-Term Irrigation with Saline Reclaimed Water”.

2, For the Abstract, detailed value in the statement like “Exogenous applications of L-alanine would increase tree size.” should be provided.

3, For the key words, mandarin should be replaced with mandarin trees.

4, For the introduction, Abbreviations and Objectives should be removed.

5, For the results, growth pictures of Mandarin trees should be exhibited.

6, For the materials and methods, genotypes of Mandarin trees employed in this study should be described. Biological and technical replicates, as well as randomization methods should be clearly stated. Plant sampling size should be included.

7, For the discussion, I would like to see the discussion section was divided into subsections with appropriate titles.

8, For the Conclusion, authors should consider to combine these several paragraphs into one paragraph.

Author Response

In this manuscript, David Auñón-Calles and colleagues examined the physiological response of mandarin trees in a semi-arid area effects over a long period of irrigation with saline reclaimed water (RW) and fresh water (FW) in terms of leaf mineral constitution, free amino acids and phytohormones balance, and their influence on yield and fruit quality. I have following comments:

1, For the title, I suggest to employ “Agronomic And Metabolic Responses Of Citrus clementina to Long-Term Irrigation with Saline Reclaimed Water”.

Reponse:

Thank you very much for the suggestion. We have modified the title and left it this way:

Agronomic And Metabolic Responses Of Citrus clementina to Long-Term Irrigation with Saline Reclaimed Water as Abiotic Factor.

2, For the Abstract, detailed value in the statement like “Exogenous applications of L-alanine would increase tree size.” should be provided.

Response:

Thank you for pointing this out. The sentence has been clarified. However, the exact value is not possible to indicate because different tests should be carry out in other Research project and, probably, such value will depend of the different conditions, treatments, crops, climatic conditions of the year, etc.

3, For the key words, mandarin should be replaced with mandarin trees.

Response:

The change has been made.

4, For the introduction, Abbreviations and Objectives should be removed.

Response:

The objectives have been removed from the text and the Abbreviations have been moved to after "Conflict of interest" according to the Instructions for Authors.

5, For the results, growth pi ctures of Mandarin trees should be exhibited.

Response:

Thank you for your suggestion. The reduction in tree size is not clearly visible in an image. However, if the reviewer deems it appropriate, we welcome the inclusion of a real image of the study trees.

6, For the materials and methods, genotypes of Mandarin trees employed in this study should be described. Biological and technical replicates, as well as randomization methods should be clearly stated. Plant sampling size should be included.

These issues are covered in the text:

The experiment was performed during 2018 in a commercial 0.5-ha orchard culti-vated with  mandarins trees (Citrus Clementina cv. ‘Orogrande’) grafted on Carrizo citrange (Citrus sinensis [L.] Osb.× Poncirus trifoliata [L.]) rootstock. All mandarins were considered adult trees. The orchard was located in Campotéjar Murcia, Spain (38◦ 07’ 18’’N; 1◦13’15’’W), where the climate is Mediterranean semi-arid with warm, dry sum-mers and mild winter conditions. Specific climate records for this experiment were ob-tained daily from a weather station near experimental orchard.

Soils within the first 90 cm depth had a loamy texture (typical according to Soil Sur-vey Staff, 2014), bulk density of 1.37 g cm-3.

A total of 96 trees, spaced at 3.5 m between plants and 5 m between rows, were used in this study. The experimental design was a randomized complete design with four blocks and four experimental plots per block. The standard plot which covered about 210 m2 was made up of twelve trees, organized in 3 adjacent rows with 4 trees per row. In each plot, the two central trees of the middle row, here after called “inner trees”, were used for measurements and the other ten trees were guard trees.

In addition, we added here a scheme with the distribution of the treatment in the field where AD is reclaimed water (RW) and AT is fresh transfer water (FW).

7, For the discussion, I would like to see the discussion section was divided into subsections with appropriate titles.

Response:

In this case, the format we chose, which is compatible with the IJMS Instructions for Authors, was to present a single Results and Discussion section rather than separate results and discussion. We have already divided this section into two general sections and the second section has been subdivided into 4 subsections.

8, For the Conclusion, authors should consider to combine these several paragraphs into one paragraph.

Response:

All paragraphs have been merged into one.

Round 2

Reviewer 1 Report

Comments and Suggestions for Authors

I have no more questions. 

Reviewer 2 Report

Comments and Suggestions for Authors

Authors corrected the manuscript according to suggestions presented in the review, manuscript is significantly improved, I have no other comments.

Reviewer 3 Report

Comments and Suggestions for Authors

Authors have addressed my concerns in the revision.